



# On the climatic influence of CO₂ forcing in the Pliocene

Lauren E. Burton[1], Alan M. Haywood[1], Julia C. Tindall[1], Aisling M. Dolan[1], Daniel J. Hill[1], Ayako Abe-Ouchi[2], Wing-Le Chan[2], Deepak Chandan[3], Ran Feng[4], Stephen J. Hunter[1], Xiangyu Li[5], W. Richard Peltier[3], Ning Tan[6], Christian Stepanek[7], Zhongshi Zhang[8]

[1]School of Earth and Environment, University of Leeds, Woodhouse Lane, Leeds, West Yorkshire, LS2 9JT, UK
[2]Atmosphere and Ocean Research Institute, The University of Tokyo, Kashiwa, 277-8564, Japan
[3]Department of Physics, University of Toronto, Toronto, M5S 1A7, Canada
[4]Department of Geosciences, College of Liberal Arts and Sciences, University of Connecticut, Storrs, CT 06033, USA
[5]Department of Atmospheric Science, School of Environmental Studies, China University of Geoscience, Wuhan, 430074, China
[6]Key Laboratory of Cenozoic Geology and Environment, Institute of Geology and Geophysics, Chinese Academy of Sciences, Beijing, 100029, China
[7]Alfred-Wegener-Institut – Helmholtz-Zentrum für Polar and Meeresforschung (AWI), Bremerhaven, 27570, Germany
[8]NORCE Norwegian Research Centre, Bjerknes Centre for Climate Research, 5007 Bergen, Norway

*Correspondence to*: Lauren E. Burton (eeleb@leeds.ac.uk)

**Abstract.** Understanding the dominant climate forcings in the Pliocene is crucial to assessing the usefulness of the Pliocene as an analogue for our warmer future. Here we implement a novel, yet simple linear factorisation method to assess the relative influence of $CO_2$ forcing in seven models of the Pliocene Model Intercomparison Project Phase 2 (PlioMIP2) ensemble. Outputs are termed "FCO₂" and show the fraction of Pliocene climate change driven by $CO_2$.

The accuracy of the FCO₂ method is first assessed through comparison to an energy balance analysis previously used to assess drivers of surface air temperature in the PlioMIP1 ensemble. After this assessment, the FCO₂ method is applied to achieve an understanding of the drivers of Pliocene sea surface temperature and precipitation for the first time.

$CO_2$ is found to be the most important forcing in the ensemble for Pliocene surface air temperature (global mean FCO₂ = 0.56), sea surface temperature (global mean FCO₂ = 0.56) and precipitation (global mean FCO₂ = 0.51). The range between individual models is found to be consistent between these three climate variables, and the models generally show good agreement on the sign of the most important forcing.

Our results provide the most spatially complete view of the drivers of Pliocene climate to date, and have implications for both data-model comparison and the use of the Pliocene as an analogue for the future. That $CO_2$ is found to be the most important forcing reinforces that the Pliocene is a good palaeoclimate analogue, but the significant effect of non-$CO_2$ forcing at regional scale reminds us that it is not perfect, and this must not be overlooked. This comparison is further complicated when considering the Pliocene as a state in quasi-equilibrium with $CO_2$ forcing compared to the transient warming being experienced at present.



# 1 Introduction

## 1.1 Pliocene climate modelling and PlioMIP

The mid-Piacenzian Warm Period (mPWP, previously referred to as the mid-Pliocene Warm Period), 3.264-3.025 Ma, is of great interest to the palaeoclimate community as a potential analogue for future climate change (Haywood et al., 2011a; Burke et al., 2018). It was the most recent period of sustained warmth above pre-industrial (PI) temperatures, is recent enough to have a continental configuration similar to modern and has similar-to-modern atmospheric $CO_2$ concentration at ~400 ppm (Pagani et al., 2010; Seki et al., 2010; Bartoli et al., 2011; de la Vega et al., 2020).

Given its potential as a palaeoclimate analogue, the study of the Pliocene has been central to palaeoclimate modelling efforts over the past three decades. In 2008, the Pliocene Model Intercomparison Project (PlioMIP) was introduced as a working group of the Palaeooclimate Model Intercomparison Project (PMIP) to further our understanding of the Pliocene climate and, in turn, its accuracy and usefulness as a palaeoclimate analogue.

PlioMIP1 focused on a climatically distinct 'time slab' spanning 3.29-2.97 Ma with temperatures generally warmer than present (Dowsett et al., 1999; 2007). PlioMIP1 comprised two experiments: seven modelling groups completed Experiment 1 with atmosphere-only climate models (Haywood et al., 2010) and eight modelling groups completed Experiment 2 with fully coupled atmosphere-ocean climate models (Haywood et al., 2011b). The large-scale feature results from PlioMIP1 were presented in Haywood et al. (2013). The ensemble showed a global mean surface air temperature (SAT) Pliocene-PI anomaly of 1.97-2.80°C and 1.84-3.60°C in Experiment 1 and Experiment 2 respectively, associated with an increase in precipitation of 0.04-0.11 mm day$^{-1}$ and 0.09-0.18 mm day$^{-1}$. Equilibrium climate sensitivity (ECS) varied between models, with an ensemble mean of 3.36°C and an Earth System Sensitivity (ESS) to ECS ratio of 1.5.

The second phase, PlioMIP2, saw the implementation of new boundary conditions in response to data-model comparison (DMC) studies of PlioMIP1 and the move from a time slab approach to a time slice focusing on a specific marine isotope stage within the mPWP with similar-to-modern orbital forcing, MIS KM5c, at 3.205 Ma. From here when we refer to the Pliocene, we are specifically referring to the MIS KM5c time slice. PlioMIP2 also saw the introduction of forcing factorisation experiments (Sect. 1.2), which allowed the influence of different climate forcings to be assessed, as well as an explicit "Pliocene4Future" element which enabled results to be directly relevant to discussions on climate sensitivity and the Pliocene as a palaeoclimate analogue (Haywood et al., 2016). 14 model groups contributed to PlioMIP2, including seven that contributed to PlioMIP1 (CCSM4, COSMOS, HadCM3, IPSLCM5A, MIROC4m, MRI-CGCM 2.3 and NorESM-L).

The large-scale feature results from PlioMIP2 were presented in Haywood et al. (2020). Global mean SAT was higher than that found in PlioMIP1, with an ensemble mean of 3.2°C warmer than the PI (range 1.7-5.2°C), partly due to the addition of models more sensitive to the Pliocene $CO_2$ forcing; the ensemble mean ECS was 3.7°C with an ESS to ECS ratio of 1.67. The increase in precipitation was also greater than seen in PlioMIP1, ranging from 0.07-0.37 mm day$^{-1}$.

The anomalies seen in PlioMIP2 are comparable to some of the Shared Socioeconomic Pathways (SSPs) shown in the Sixth Assessment Report of the Intergovernmental Panel on Climate Change (IPCC AR6; Fig. 1), reinforcing the potential to use



the Pliocene as a palaeoclimate analogue. The magnitude of global mean warming relative to the PI is comparable between
the Pliocene (3.2°C; Haywood et al., 2020) and end of the century (2081-2100) estimates for SSP2-4.5 (2.7°C) and SSP3-7.0
(3.6°C; Lee et al., 2021), though the latter may look even more comparable to the Eocene (Burke et al., 2018; Lee et al., 2021).

70 There are also comparable spatial patterns of climate anomalies between end of the century and PlioMIP2 in the form of polar
amplification and the land warming more than the ocean (Fig. 1a, b, c).



**Figure 1: PlioMIP2 ensemble MIS KM5c SAT (a) and precipitation (d) anomalies relative to the PI compared to equivalent CMIP6 anomalies for 2081-2100 under SSP2-4.5 (c, d) and SSP3-7.0 (e, f). The PlioMIP2 ensemble includes all 16 models in Haywood et al. (2020) plus HadGEM3 (Williams et al., 2021). The CMIP6 SAT anomalies (c, d) are relative to 1850-1900, and precipitation anomalies (e, f) to the standard CMIP6 base period (1995-2014; Gutiérrez et al., 2021).**

From the water cycle projections in the IPCC AR6 (Table 8.1 in Douville et al., 2021), it is clear that the global mean percentage change in precipitation is also comparable between 2081-2100 under SSP2-4.5 (4.0%) and SSP3-7.0 (5.1%) relative to the CMIP6 base period (1995-2014) and the Pliocene (7%; Haywood et al., 2020). Similar spatial features include the wetting of the Sahara and polar regions, and drying of the Caribbean, off the western coast of South America (Fig. 1d, e, f).

However, caution must be applied when referencing the Pliocene as a palaeoclimate analogue given the importance of – continually changing – anthropogenic greenhouse gas forcing in present day.

Here, we begin to assess the role of $CO_2$ forcing in the Pliocene compared to other drivers of climate and changes in boundary conditions. The non-$CO_2$ forcing we refer to includes changes to ice sheets and 'orography', the latter of which also includes changes to prescribed vegetation, bathymetry, land-sea mask, soils and lakes per the experimental design of PlioMIP2 (Haywood et al., 2016).

## 1.2. Drivers of Pliocene climate

Though there are similarities in large-scale climate features between the Pliocene and end-of-century projections in AR6, the similarity in the causes and drivers of some of these features is yet to be fully assessed.

Previous studies on the drivers of Pliocene temperature change have used energy balance analyses. These are commonly applied in palaeoclimate studies to understand changes in temperature by separating out individual forcing components (e.g. Lunt et al., 2012; Hill et al., 2014 and references therein).

Lunt et al. (2012) combined a novel factorisation methodology with energy balance analysis to assess the causes of Pliocene warmth in HadCM3 using the PRISM2 boundary conditions (Dowsett et al., 1999). $CO_2$ was found to cause 36-61% of Pliocene warmth, orography 0-26%, ice sheets 9-13% and vegetation 21-27%. These drivers were found to have spatial variation in importance, with changes in orography and ice sheets particularly important in driving polar amplification in the northern high-latitudes, and orography particularly important in the southern high-latitudes. The energy balance analysis also highlighted how surface albedo changes and direct $CO_2$ forcing contributed more than cloud feedbacks, with surface albedo changes dominating at mid- and high-latitudes and $CO_2$ forcing at low latitudes.

Hill et al. (2014) developed the methodology of Lunt et al. (2012) and conducted the first multi-model energy balance analysis using the eight models included in PlioMIP1 Experiment 2, forced with the PRISM3 boundary conditions. Greenhouse gas emissivity was found to be the dominant cause of warming in the tropics. There were large uncertainties between models in the high latitudes but all energy balance components were important, and clear sky albedo was the dominant driver of polar amplification through reductions in ice sheets, sea ice and snow cover and changes to vegetation. The relative influence of the energy balance components was more uncertain in the northern mid-latitudes, particularly in the North Atlantic and Kuroshio Current regions, where warming was also simulated differently between models (Haywood et al., 2013).





Developing from PlioMIP1, forcing factorisation experiments were included in PlioMIP2 to enable the explicit assessment of forcing components (Haywood et al., 2016). These experiments included Pliocene simulations with PI ice configuration (experiment $Eo^{400}$) and PI orography configuration (experiment $Ei^{400}$), as well as a PI simulation with Pliocene-level $CO_2$

concentration (experiment $E^{400}$); the PlioMIP2 experimental design and naming conventions were shown in Haywood et al. (2016). These forcing factorisation experiments were in Tier 2 of the experimental design, meaning they were optional and completed by a smaller number of model groups.

The impact of various mechanisms on Pliocene climate has been studied using energy balance analysis in individual PlioMIP2 models. Using the PlioMIP2 forcing factorisation experiments and methodology proposed in Haywood et al. (2016), Chandan

and Peltier (2018) assessed the mechanisms of Pliocene climate in the CCSM4-UoT model. They found that around 1.67°C (45%) of warming was attributable to $CO_2$ forcing, 1.54°C (42%) to changes in orography and 0.47°C (13%) to a reduction in ice sheets. Using the same factorisation methodology for the COSMOS model, Stepanek et al. (2020) found that 2.23°C (~66%) of warming was attributable to $CO_2$ forcing, 0.91°C (~25%) to orography and 0.38°C (~13%) to changes in the ice sheets.

An updated methodology of Lunt et al. (2012) and Hill et al. (2014) is used to explore drivers of northern high-latitude warmth

in the CCSM4 model in Feng et al. (2017). Changes to regional topography, Arctic sea ice and the Greenland ice sheet, and the North Atlantic meridional overturning circulation were found to explain the amplification of SAT in the northern high-latitudes. Greenhouse gas emissivity was also found to be important, particularly with the subsequent positive feedbacks which have a more distributed effect. This updated methodology is also used in Feng et al. (2019) where it is demonstrated that a seasonally sea ice-free Pliocene Arctic Ocean can be simulated in CESM1.2 by including aerosol-cloud interactions and

excluding industrial pollutants.

To date, there has been no systematic study comparing multiple models in the PlioMIP2 ensemble to spatially quantify the importance of different climate forcings, nor have climate variables other than SAT been previously assessed in multiple models in a single study. Here, we present the relative spatial influence of $CO_2$ forcing for SAT across multiple PlioMIP2 models and, for the first time, sea surface temperature (SST) and precipitation. We employ the forcing factorisation

experiments of PlioMIP2 and a novel, simple linear factorisation method with outputs we term "$FCO_2$".

## 2. Methods

### 2.1. Model boundary conditions

Standardised boundary conditions are used by all model groups for the core Pliocene control experiment in PlioMIP2, derived from the U.S. Geological Survey PRISM4 reconstruction (Dowsett et al., 2016) and implemented as described in Haywood et

al. (2016). These boundary conditions include spatially-complete gridded datasets at 1° x 1° of latitude-longitude for land-sea distribution, topography and bathymetry, vegetation, soil, lakes and land ice cover; all models analysed here use the "enhanced" version of the boundary conditions, meaning they include all reconstructed changes to the land-sea mask and ocean bathymetry (Haywood et al., 2020).





The configuration of the Greenland ice sheet in PRISM4 is based upon the results from the Pliocene Ice Sheet Modelling
Intercomparison Project (PLISMIP): it is confined to high elevations in the Eastern Greenland Mountains and covers an area
around 25% of the modern ice sheet (Dolan et al., 2015; Koenig et al., 2015). Ice coverage over Antarctica has been debated
(see Levy et al., 2022) but the PRISM3 Antarctic ice configuration – in which there is a reduction in the ice margins in the
Wilkes and Aurora basins in eastern Antarctica, and western Antarctica is largely ice free (Dowsett et al., 2010) – is supported
and so retained in the PRISM4 reconstruction (Dowsett et al., 2016). Later modelling studies further support the potential for
ice retreat in similar areas in Antarctica under the warmer temperatures of the Pliocene (e.g. DeConto and Pollard, 2016).
The palaeogeography is broadly similar to modern except for the closure of the Bering Strait and Canadian Arctic Archipelago;
changes in the Torres Strait, Java Sea, South China Sea, Kara Strait; and a West Antarctic Seaway (Haywood et al., 2016).
PRISM4 also includes dynamic topography and glacial isostatic adjustment for the first time to inform the representation of
local sea level (Dowsett et al., 2016).
Atmospheric $CO_2$ concentration is set to 400 ppm and, in the absence of proxy data, all other trace gases are set to be identical
to the concentrations in the PI control experiment for each individual model group (Haywood et al., 2016).

## 2.2. Participating models

Seven of the 17 models of the PlioMIP2 ensemble are included in this study as they conducted the necessary experiments to
apply our novel $FCO_2$ method: $Eoi^{400}$, $E^{400}$ and $E^{280}$ (see Sect. 2.3). This subgroup is also found to be representative of the
wider PlioMIP2 ensemble in terms of modelled ECS, ESS and global mean $Eoi^{400}$-$E^{280}$ SAT anomaly (Table 1).

| Parameter | PlioMIP2 | This ensemble |
|---|---|---|
| ECS (°C) | 3.7 | 3.8 |
| ESS (°C) | 6.2 | 6.1 |
| ESS to ECS ratio | 1.7 | 1.7 |
| $Eoi^{400}$-$E^{280}$ SAT anomaly (°C) | 3.2 | 3.2 |

**Table 1: A comparison of climate parameters between the PlioMIP2 ensemble and the subgroup of PlioMIP2 models used here.**

The models are of varying ages and resolutions. Summary details relevant to PlioMIP2 are shown in Haywood et al. (2020)
and in individual model papers for the PlioMIP2 experiments which are cited in Table 2.

| Model | Vintage | Sponsor(s), country | $Eoi^{400}$ boundary conditions and experiment citation | Climate sensitivity (ECS; °C) and citation |
|---|---|---|---|---|
| CCSM4-UoT | 2011 | University of Toronto, Canada | Enhanced with fixed vegetation (Chandan and Peltier, 2017; 2018) | 3.2 (Peltier and Vettoretti, 2014; Chandan and Peltier, 2018) |
| CESM2 | 2020 | National Center for Atmospheric Research, USA | Enhanced with fixed vegetation (Feng et al., 2020) | 5.3 (Gettelman et al., 2019) |





| COSMOS | 2009 | Alfred Wegener Institute, Germany | Enhanced with dynamic vegetation (Stepanek et al., 2020) | 4.7 (Stepanek et al., 2020) |
|---|---|---|---|---|
| HadCM3 | 1997 | University of Leeds, UK | Enhanced with fixed vegetation (Hunter et al., 2019) | 3.5 (Hunter et al., 2019) |
| IPSLCM5A2 | 2017 | LSCE, France | Enhanced with fixed vegetation (Tan et al., 2020) | 3.6 (reported in Haywood et al., 2020) |
| MIROC4m | 2004 | Center for Climate Research (Uni. Tokyo, National Inst. For Env. Studies, Frontier Research Center for Global Change, JAMSTEC), Japan | Enhanced with fixed vegetation (Chan and Abe-Ouchi, 2020) | 3.9 (Chan and Abe-Ouchi, 2020) |
| NorESM1-F | 2017 | NORCE Norwegian Research Centre, Bjerknes Centre for Climate Research, Bergen, Norway | Enhanced with fixed vegetation (Li et al., 2020) | 2.3 (Guo et al., 2019) |

**Table 2: Details of the climate models used in the FCO₂ analysis (adapted from Haywood et al., 2020).**

## 2.3. FCO₂ method

Taking advantage of the forcing factorisation experiments included in the PlioMIP2 experimental design, here we propose a novel simple linear factorisation method to assess the influence of $CO_2$ forcing with outputs we term "FCO₂". We apply the FCO₂ method to all seven models for SAT and precipitation, and to six models for SST; IPSLCM5A2 is excluded for analysis of the latter as only 10 model years of data were available.

The method uses three PlioMIP2 experiments: the two core experiments ($E^{280}$ and $Eoi^{400}$) and one Tier 2 experiment ($E^{400}$; Table 3). Core experiments were completed by all PlioMIP2 modelling groups and Tier 2 experiments were submitted by a smaller number of modelling groups. The seven models included here were the only ones to have reported $E^{400}$ results by the time of compiling this study.

| ID | Description | Land-sea mask | Topography | Ice | Vegetation | CO₂ (ppm) | Status |
|---|---|---|---|---|---|---|---|
| $Eoi^{400}$ | Pliocene control experiment | Pliocene - Modern | Pliocene | Pliocene | Dynamic | 400 | Core |
| $E^{280}$ | PI control | Modern | Modern | Modern | Dynamic | 280 | Core |
| $E^{400}$ | PI experiment with CO₂ concentration of 400 ppm | Modern | Modern | Modern | Dynamic | 400 | Tier 2 – Pliocene4Future and Pliocene4Pliocene |

**Table 3: Details of the PlioMIP2 experiments included in the FCO₂ analysis (adapted from Haywood et al., 2016). Note that dynamic vegetation was optional in the experimental design: only COSMOS ran with dynamic vegetation and all other models ran with the**





**prescribed vegetation of Salzmann et al. (2008). As COSMOS ran with dynamic vegetation, some vegetation feedback in this model will be included in the $E^{400}$-$E^{280}$ anomaly.**

We define $FCO_2$ as an approximation of the relative influence of $CO_2$ calculated by:

$$FCO_2 = \frac{(E^{400} - E^{280})}{(Eoi^{400} - E^{280})} \tag{1}$$

where $E^{400}$-$E^{280}$ represents the change in climate caused by the change in $CO_2$ concentration from 280 ppm to 400 ppm alone, and $Eoi^{400}$-$E^{280}$ represents the change in climate as a result of implementing the full Pliocene boundary conditions.

$FCO_2$ is therefore a fractional quantity where a value of 1.0 denotes that the signal of change is wholly dominated by $CO_2$ forcing, and a value of 0.0 denotes the contrasting case where the climate signal wholly is dominated by non-$CO_2$ forcing. Here, non-$CO_2$ forcing is defined as changes to ice sheets and orography, the latter of which includes changes to prescribed

vegetation, bathymetry, land-sea mask, soils and lakes per the PlioMIP2 experimental design (Haywood et al., 2016). Our full interpretation of the range of $FCO_2$ values is shown in Table 4.

| $FCO_2$ value | Interpretation relative to signal of change |
|---|---|
| >1.0 | Wholly dominated by $CO_2$ forcing with some non-$CO_2$ forcing acting in the opposite direction to the overall climate signal |
| 0.8-1.0 | Highly dominated by $CO_2$ forcing |
| 0.6-0.8 | Dominated by $CO_2$ forcing |
| 0.5-0.6 | Mixed forcing but $CO_2$ forcing dominant |
| 0.4-0.5 | Mixed forcing but non-$CO_2$ forcing dominant |
| 0.2-0.4 | Dominated by non-$CO_2$ forcing |
| 0.0-0.2 | Highly dominated by non-$CO_2$ forcing |
| <0.0 | Wholly dominated by non-$CO_2$ forcing with some $CO_2$ forcing acting in the opposite direction to the overall climate signal |

**Table 4: Interpretation of $FCO_2$ values.**

$FCO_2$ values are not limited between 0.0 and 1.0. $FCO_2$ values above 1.0 represent a climate that is wholly dominated by $CO_2$ forcing where non-$CO_2$ forcing creates an opposing climatic effect. Similarly, $FCO_2$ values below 0.0 represent a climate that

is wholly dominated by non-$CO_2$ forcing where $CO_2$ forcing creates an opposing climatic effect.

This becomes clear if one considers $FCO_2$ in the case of SAT and SST. The Pliocene climate is characterised as having elevated temperature and $CO_2$ concentration compared to the PI (e.g. Dowsett et al., 2016; Haywood et al., 2020) so, given that the predominant effect of $CO_2$ forcing is warming, an $FCO_2$ value below 0.0 is rare. An exception is provided by higher-order effects where $CO_2$ leads to a cooling (see Sherwood et al., 2020). $FCO_2$ values below 0.0 for SAT are limited to central

Antarctica where the overall Pliocene climate change is a cooling with respect to the PI (see Sect. 3.2), and there are no $FCO_2$ values below 0.0 for SST.



We consider uncertainty in the $FCO_2$ method in terms of whether there is consistent agreement between the individual models on whether $CO_2$ forcing or non-$CO_2$ forcing is the most important driver (i.e. whether $FCO_2 > 0.5$ or $FCO_2 < 0.5$). In this paper, we deem $FCO_2$ to be "uncertain" if four or fewer models agree on the dominant forcing (see Fig. 7, Sect. 4.1).

In checking for non-linearity, we consider an additional PlioMIP2 simulation that tests the effect of Pliocene boundary conditions with PI-level $CO_2$ concentration ($Eoi^{280}$). The sum contributions of $CO_2$ and non-$CO_2$ factors relative to the total $Eoi^{400}$-$E^{280}$ anomaly is close to zero (Eq. 2; Supplementary Fig. 1 (S1)), meaning that any factors not considered in these experiments – i.e. anything other than $CO_2$ concentration, changes to ice sheets, orography and/or vegetation – are unlikely to be a dominant cause of change. Non-linearity is tested for in the four models which had reported $Eoi^{280}$ results by the time of

compiling this study: CCSM4-UoT, COSMOS, HadCM3 and MIROC4m. Additional checks with the other models would likely further confirm the linearity, and highlights the utility in more modelling groups completing the forcing factorisation experiments in PlioMIP2 and future phases.

$$(Eoi^{400} - E^{280}) - [(E^{400} - E^{280}) + (Eoi^{280} - E^{280})] \approx 0 \tag{2}$$

### 2.4. Energy balance analysis

Results from the $FCO_2$ method are compared to an energy balance analysis using the methodology of Hill et al. (2014). This methodology was developed from the factorisation methodology of Heinemann et al. (2009) and Lunt et al. (2012) and assumed that the change in SAT was largely driven by $CO_2$, orography, ice sheets and vegetation and that any other changes (such as soils or lakes) had a negligible impact:

$$\Delta T = dT_{CO_2} + dT_{orog} + dT_{ice} + dT_{veg} \tag{3}$$

In the Hill et al. (2014) methodology, the temperature at each latitude in a GCM experiment is given by:

$$T = \left(\frac{SW^{\downarrow}_{TOA} - (1-\alpha) - H}{\varepsilon \sigma}\right)^{1/4} \equiv T(\varepsilon, \alpha, H) \tag{4}$$

with the temperature anomaly approximated by a linear combination of contributions from changes in emissivity ($\Delta T_\varepsilon$), albedo ($\Delta T_\alpha$) and heat transport ($\Delta T_H$). Temperature changes due to emissivity and albedo can be further separated to include changes attributed to the impact of atmospheric greenhouse gases ($\Delta T_{gg\varepsilon}$), clouds (through impacts on emissivity, $\Delta T_{c\varepsilon}$, and albedo,

$\Delta T_{c\alpha}$), and clear sky albedo ($\Delta T_{cs\alpha}$). The effect of changes in temperature due to topography ($\Delta T_{topo}$) is also important to consider when comparing the Pliocene to the PI where specific details differ (Dowsett et al., 2016).

$$\Delta T \approx \Delta T_{gg\varepsilon} + \Delta T_{c\varepsilon} + \Delta T_{c\alpha} + \Delta T_{cs\alpha} + \Delta T_H + \Delta T_{topo} \tag{5}$$

where

$$\Delta T_{gg\varepsilon} = T(\varepsilon_{cs}, \alpha, H) - T(\varepsilon'_{cs}, \alpha, H) - \Delta T_{topo} \tag{6}$$

and

$$\Delta T_{topo} = \Delta h \cdot \gamma \tag{7}$$





in which $\Delta h$ is the change in topography (Pliocene-PI) and $\gamma$ is a constant atmospheric lapse rate ($\approx$5.5 K km$^{-1}$, Yang and Smith, 1985; Hill et al., 2014).

This more approximate methodology is chosen over the further modified methodology of Feng et al. (2017) – in which an
amended approximate partial radiative perturbation method was applied to calculate cloud-sky albedo more accurately in polar regions, and zonal heat transport was separated into atmosphere and ocean components – as it was used to assess the PlioMIP1 ensemble thus provides directly comparable outputs.

Six of the seven models for which $FCO_2$ is quantified are considered in the energy balance analysis; IPSLCM5A2 is excluded because the required fields were not available for this model. Model-specific topography files are used as they were
implemented in the individual model E$^{280}$ and Eoi$^{400}$ experiments to minimise uncertainties that may arise due to different implementation methods, and the energy balance components are compared to the simulated temperature change and outputs of the $FCO_2$ method. The multi-model mean (MMM) energy balance is calculated using the MMM of each of the individual components:

$$\overline{\Delta T} \approx \overline{\Delta T_{gg\varepsilon}} + \overline{\Delta T_{c\varepsilon}} + \overline{\Delta T_{c\alpha}} + \overline{\Delta T_{cs\alpha}} + \overline{\Delta T_{H}} + \overline{\Delta T_{topo}} \tag{8}$$

Comparing the SAT outputs of the energy balance analysis with outputs of the $FCO_2$ method on SAT allow the accuracy of our novel method to be assessed, and also aids the interpretation of, and adds nuance to, the $FCO_2$ results. In order to assess the accuracy of the simple linear estimate and to further validate the $FCO_2$ method, we compare $\Delta T_{gg\varepsilon}$ to the E$^{400}$-E$^{280}$ SAT anomaly, and the sum of $\Delta T_{c\varepsilon}$, $\Delta T_{c\alpha}$, $\Delta T_{cs\alpha}$, $\Delta T_{H}$ and $\Delta T_{topo}$ to the Eoi$^{400}$-E$^{400}$ SAT anomaly.

## 3. Results

### 3.1. Energy balance analysis

The Eoi$^{400}$-E$^{280}$ energy balance analysis unravels the relative contributions of $CO_2$, topography, cloud emissivity, clear sky albedo and heat transport to the Eoi$^{400}$-E$^{280}$ SAT anomaly (Fig. 2). The energy balance analysis for the sub-group of PlioMIP2 models presented here supports the findings of the PlioMIP1 ensemble presented in Hill et al. (2014): clear sky albedo is the dominant driver of warming and polar amplification in the high-latitudes, and greenhouse gas emissivity is the dominant driver
in the low-latitudes. The zonal influence of $CO_2$ on Pliocene warming also appears relatively consistent across latitudes as in Hill et al. (2014); there is some amplification at high-latitudes, particularly the Northern Hemisphere, but this amplification is smaller than that seen for other energy balance components.



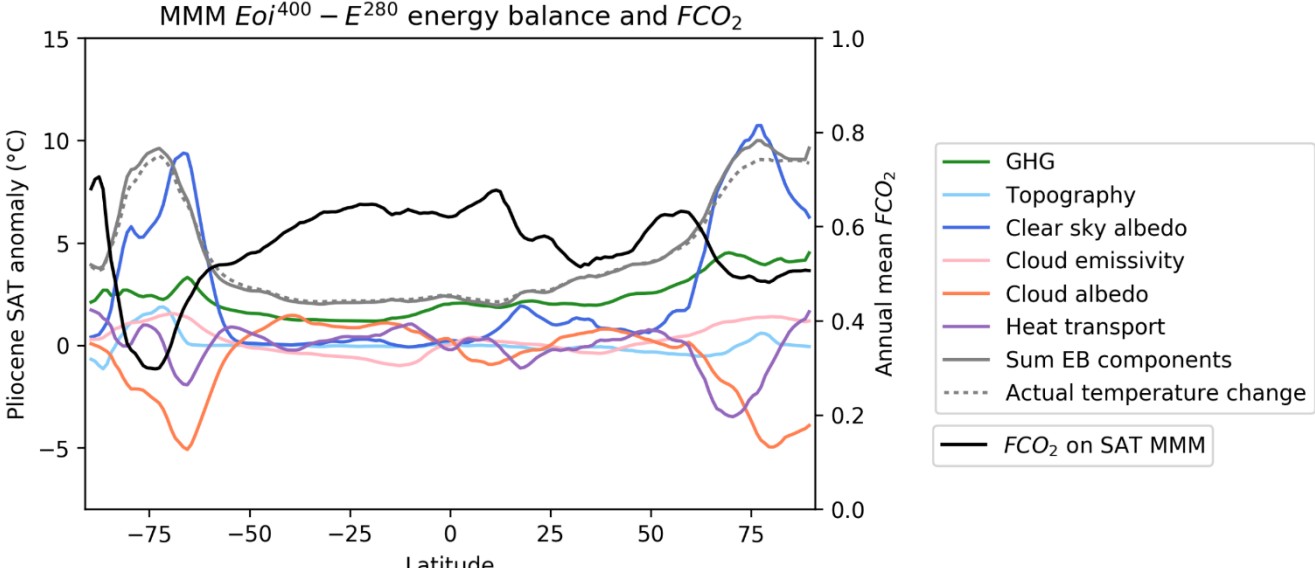

**Figure 2: The MMM Eoi[400]-E[280] energy balance with the $FCO_2$ of the SAT MMM. The MMM includes HadCM3, COSMOS,**
**CCSM4-UoT, CESM2, MIROC4m and NorESM1-F for both the energy balance and $FCO_2$ (IPSLCM5A2 is excluded because the required fields were not available for the energy balance analysis). The degree of Pliocene warming attributable to each energy balance component at each degree latitude is shown and the sum of the energy balance terms (solid grey line) agrees well with the simulated temperature change (dashed grey line). The $FCO_2$ of the SAT MMM is shown in the solid black line with a separate axis to compare to the energy balance.**

The $FCO_2$ method provides an alternative estimate for the relative contribution of $CO_2$ to changes in SAT compared to the energy balance analysis. $FCO_2$ is lower than the greenhouse gas contribution as computed in the energy balance analysis at the high-latitudes (with the exception of the very high latitudes in the Southern Hemisphere) where there is a greater contribution from clear sky albedo and topography. Conversely, it is higher in the mid- and low-latitudes where $CO_2$ is the dominant energy balance component.

The energy balance analysis provides more nuance on the specific drivers of change than the $FCO_2$ method, which only indicates whether warming is due to $CO_2$ forcing or non-$CO_2$ forcing. Using the energy balance analysis in tandem with $FCO_2$, we are able to understand which component/s within the encompassing "non-$CO_2$" category is/are most influential. For example, the energy balance analysis highlights how clear sky albedo has the largest influence on Pliocene warming in the high-latitudes, where the $FCO_2$ method suggests that non-$CO_2$ factors are important. Furthermore, the energy balance analysis

helps to explain the reasons for $FCO_2$ values above 1 for SAT; for example, although only at a zonal scale, the energy balance analysis shows that topography acts to lower SAT at the South Pole. However, the $FCO_2$ method provides spatial nuance not possible with the energy balance analysis (see Sect. 4.1).

The energy balance analysis can also be compared with the E[400]-E[280] and Eoi[400]-E[400] SAT anomalies (Fig. 3). The greenhouse gas energy balance component ($\Delta T_{gg\epsilon}$) is seen to be in good agreement with the E[400]-E[280] SAT anomaly (Fig. 3a), with a global





mean increase in SAT of 1.97°C and 1.85°C respectively. The energy balance component shows more variability and uncertainty between models than the E$^{400}$-E$^{280}$ anomaly and also shows more zonal variation.

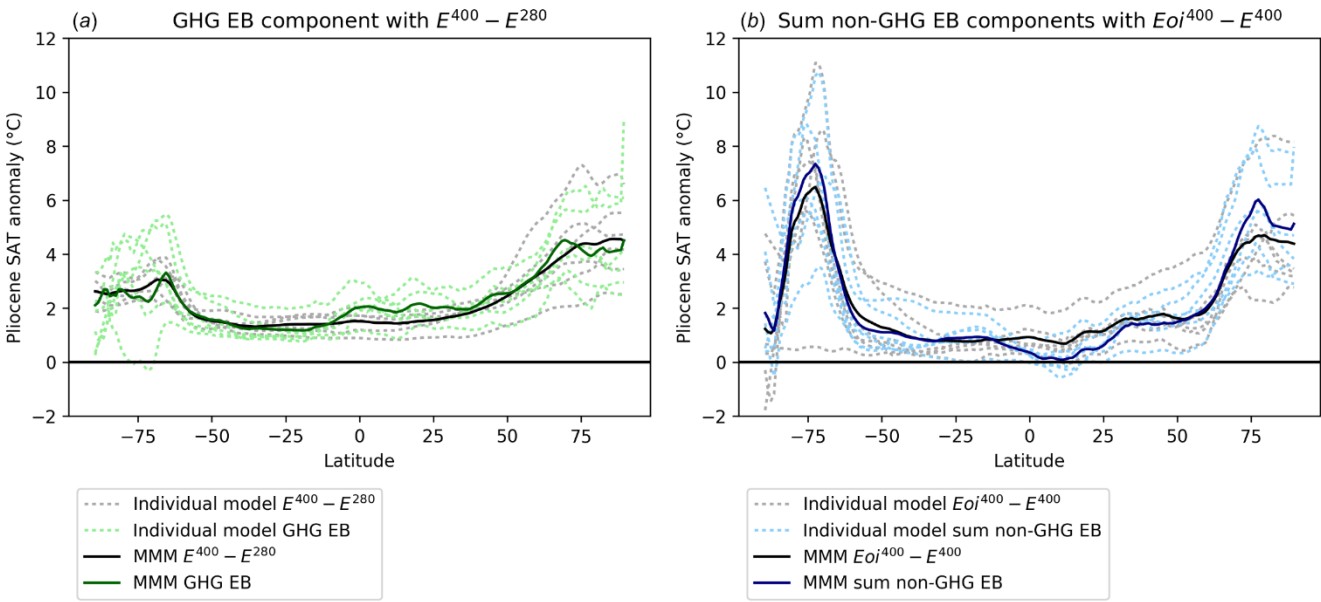

**Figure 3: Comparison of the greenhouse gas (GHG) energy balance component ($\Delta T_{gg\varepsilon}$) and the E$^{400}$-E$^{280}$ SAT anomaly (a) and equivalent comparison of the sum of non-greenhouse gas energy balance components and the Eoi$^{400}$-E$^{400}$ anomaly (b). The MMM is**
**shown in a solid line and individual models by dotted lines, representing uncertainty between models.**

The sum of the non-greenhouse gas energy balance components is also seen to be in good agreement with the Eoi$^{400}$-E$^{400}$ anomaly (Fig. 3b), with a global mean increase in SAT of 1.38°C and 1.49°C respectively. There is more uncertainty between models for the Eoi$^{400}$-E$^{400}$ anomaly, highlighting the different implementations of ice sheets and land-sea masks in the Eoi$^{400}$ experiment.

That the absolute anomalies and energy balance components agree provides an additional argument for the accuracy and usefulness of the simple linear estimations used in the FCO$_2$ method, and hence enables the first estimates of the drivers of SST (Sect. 3.3) and precipitation (Sect. 3.4), as well as more spatially detailed estimates of the drivers of SAT (Sect. 3.2).

**3.2. Surface air temperature**

The MMM Eoi$^{400}$-E$^{280}$ global mean SAT anomaly is 3.2°C, equal to the anomaly of the PlioMIP2 ensemble (Haywood et al.,
2020). The range is also equal to the PlioMIP2 ensemble, with end members NorESM1-F and CESM2 simulating the smallest (1.7°C) and largest (5.2°C) Eoi$^{400}$-E$^{280}$ anomalies respectively (Haywood et al., 2020). Warming occurs in all regions and is amplified in the high-latitudes, except for an isolated region of cooling in central Antarctica (Fig. 4a).




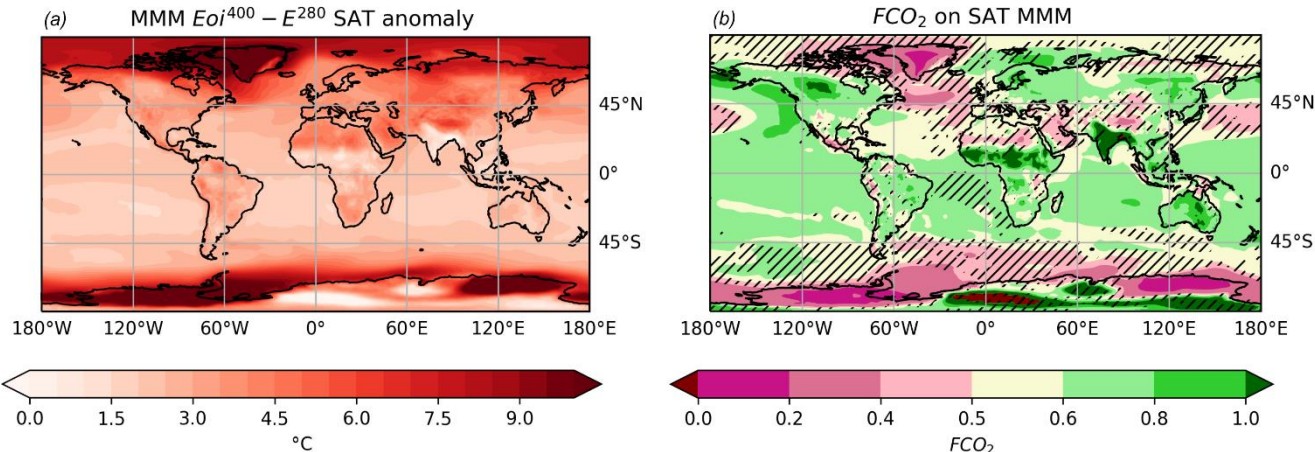

**Figure 4: MMM Eoi$^{400}$ – E$^{280}$ SAT anomaly (a) and FCO$_2$ of the MMM for SAT (b). Hatching in (b) represents where there is no**
**consistent agreement between models on whether CO$_2$ forcing or non-CO$_2$ forcing is the most important (i.e. whether FCO$_2$ > 0.5 or**
**FCO$_2$ < 0.5).**

The MMM global mean FCO$_2$ is 0.56 (individual model range 0.40-0.70; Fig. S2), meaning 56% of the SAT change is due to
CO$_2$ forcing. FCO$_2$ varies around the globe (Fig. 4b): CO$_2$ is the most important forcing in large areas of the low latitudes and,
predictably, becomes less important in the high latitudes due to the significant changes in ice sheets and orography in the
Pliocene. FCO$_2$ is found to be similar over land and ocean, with mean values of 0.58 and 0.56 respectively.

Many areas of highly dominant CO$_2$ forcing (FCO$_2$ 0.8-1.0) are found on land: over central Africa, the Indian subcontinent,
and parts of Australia, Antarctica and North America. Parts of these areas have FCO$_2$ values above 1.0, indicating where non-
CO$_2$ forcing acts in the opposite direction to the overall signal. However, high FCO$_2$ is also seen in the Pacific Ocean off the
western coast of North America and the Barents Sea south of Svalbard.

There is a small region in the North Atlantic off the eastern coast of North America where non-CO$_2$ forcing is dominant (FCO$_2$
0.2-0.4) but regions where non-CO$_2$ forcing is highly dominant (FCO$_2$ 0.0-0.2) are mostly limited to Antarctica and Greenland,
evidencing the role of changes to orography and ice sheets in polar amplification in the Pliocene. In central Antarctica there is
also a region where FCO$_2$ is below 0.0, indicating where CO$_2$ forcing is acting to warm the climate against an overall signal of
cooling.

The FCO$_2$ method shows CO$_2$ to be the most important forcing overall but there is also a significant contribution from non-
CO$_2$ forcing which should not be overlooked, particularly if we are to learn from the Pliocene as an analogue for the future.
Regions of uncertainty are generally found where the dominant forcing is mixed (FCO$_2$ 0.4-0.6) but there is also uncertainty
in some regions of dominant and highly dominant CO$_2$ forcing (FCO$_2$ 0.6-1.0), including central and eastern Antarctica, the
Barents Sea and isolated regions of central Africa and of the Indian subcontinent (Fig. 4b). Other notable regions of uncertainty
include the North Atlantic and northwest Pacific, consistent with the findings of Hill et al. (2014); however, in our analysis of
PlioMIP2 simulations we find that the northern mid-latitudes appear to have more certainty than in the PlioMIP1 ensemble.





### 3.3. Sea surface temperature

The MMM Eoi[400]-E[280] global mean SST anomaly is 2.3°C, again equal to the global mean anomaly of the PlioMIP2 ensemble.

The anomaly also sits relatively central to the PlioMIP2 ensemble range of 1.3-3.9°C (Haywood et al., 2020). Warming is seen

in all ocean basins with amplification in the high-latitudes, particularly in the Labrador Sea and North Atlantic (Fig. 5a).

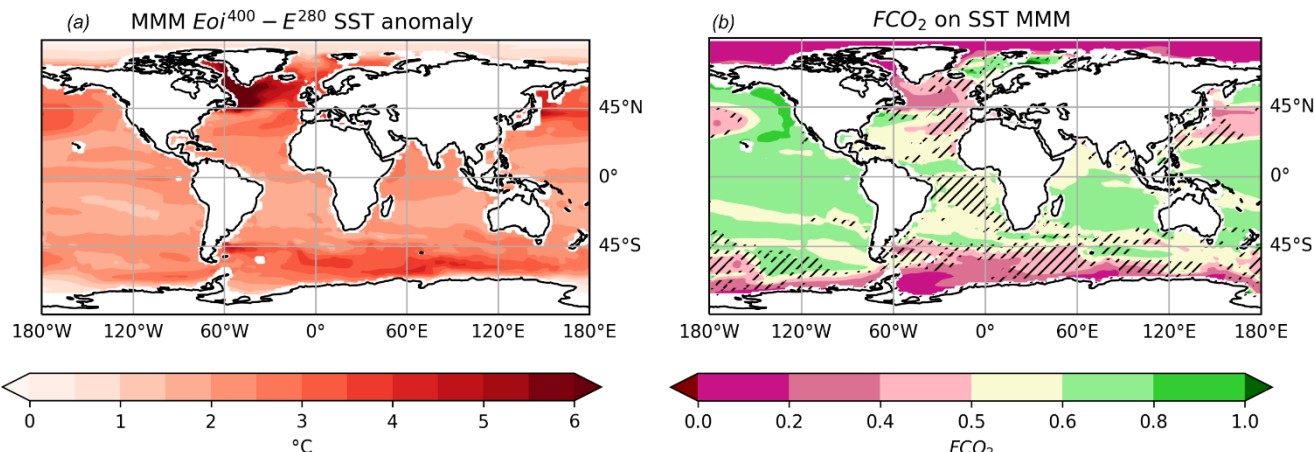

**Figure 5: MMM Eoi[400] – E[280] SST anomaly (a) and FCO₂ of the MMM for SST (b). IPSLCM5A2 is excluded from SST analysis due to limited data availability. Hatching in (b) represents where there is no consistent agreement between models on whether CO₂ forcing or non-CO₂ forcing is the most important (i.e. whether FCO₂ > 0.5 or FCO₂ < 0.5).**

The MMM global mean $FCO_2$ is 0.56 (individual model range 0.40-0.76; Fig. S3), meaning 56% of the SST change is due to $CO_2$ forcing. The MMM global mean $FCO_2$ on SST is the same as the MMM global mean $FCO_2$ on SAT and there are comparable spatial features at low- and mid-latitudes. On the other hand, $FCO_2$ on SST is significantly lower than on SAT at high-latitudes (Fig. 5b), indicating where changes in orography and ice sheets, and feedbacks including sea ice, have a much larger influence on SST than they have on SAT.

Non-$CO_2$ forcing is dominant or highly dominant ($FCO_2$ 0.0-0.4) in the Arctic Sea, and dominant in much of the Southern Ocean. SST in the South Atlantic is also more strongly driven by non-$CO_2$ forcing than for SAT in the region, perhaps indicating a change in ocean circulation driven by these non-$CO_2$ forcings consistent with previous work (e.g. Hill et al., 2017). No regions of $FCO_2$ below 0.0 are seen.

The amplified warming seen in the Labrador Sea and North Atlantic appears to be predominantly driven by non-$CO_2$ forcing

($FCO_2$ 0.2-0.4), but the warming pattern also extends to regions where forcing is mixed ($FCO_2$ 0.4-0.6) or, south of Svalbard, where forcing is even dominated by $CO_2$ ($FCO_2$ 0.6-0.8).

Regions of uncertainty in $FCO_2$ on SST largely mirror those for SAT over the sea surface and are predominantly found in regions of mixed forcing ($FCO_2$ 0.4-0.6), and in the mid- and southern high-latitudes. Unlike for SAT, SSTs in the Arctic Ocean show good agreement that non-$CO_2$ forcing is highly dominant ($FCO_2$ 0.0-0.2). This difference in consistency between

$FCO_2$ on SAT and $FCO_2$ on SST might relate to the different distributions of sea ice between models.



### 3.4. Precipitation

The MMM $Eoi^{400}$-$E^{280}$ precipitation anomaly is 0.18 mm day$^{-1}$ or 6.4% compared to the PlioMIP2 ensemble value of 7% (range 2-13%; Haywood et al., 2020).

Particularly large increases in precipitation are seen in northern Africa and the Middle East, as well as over Greenland and
parts of Antarctica (Fig. 6a). The MMM spatial pattern of precipitation change is more complex than that seen for SAT and SST but is representative of the whole PlioMIP2 ensemble (Fig. 6a; Fig. 1d).

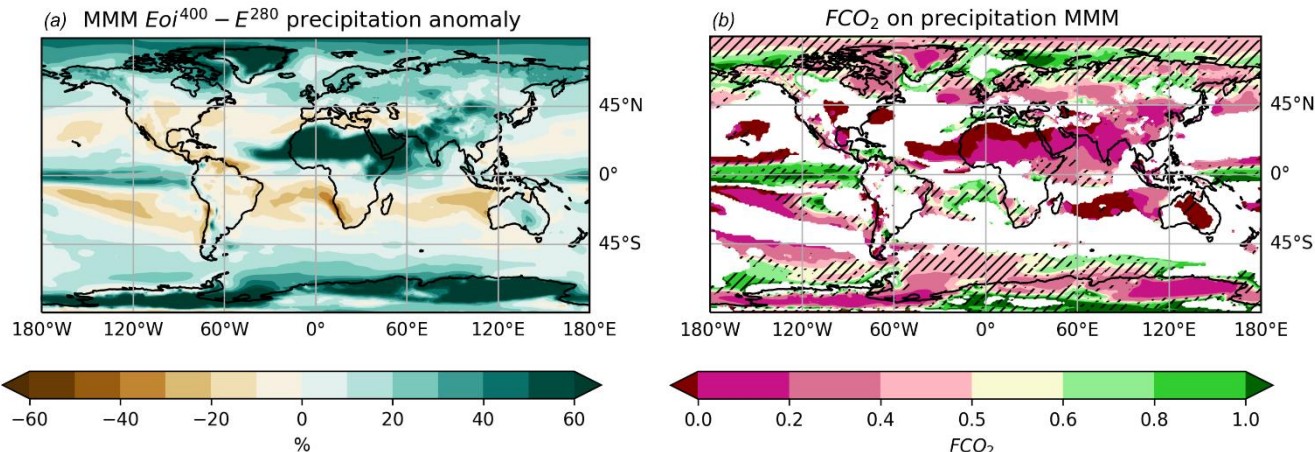

**Figure 6: MMM $Eoi^{400}$ – $E^{280}$ precipitation anomaly (a) and $FCO_2$ of MMM for precipitation (b). In (b), regions of $Eoi^{400}$-$E^{280}$ precipitation change less than 10% are masked (white) and hatching represents where there is no consistent agreement between**
**models on whether $CO_2$ forcing or non-$CO_2$ forcing is the most important (i.e. whether $FCO_2 > 0.5$ or $FCO_2 < 0.5$).**

The spatial pattern of $FCO_2$ on precipitation is also more complex than that seen for SAT and SST; areas of percentage change less than 10% are masked in white to increase clarity and reduce noise (Fig. 6b). The MMM global mean $FCO_2$ is 0.51 (individual model range 0.39-0.69; Fig. S4), meaning $CO_2$ forcing causes 51% of the change in global mean precipitation and so non-$CO_2$ forcing plays a slightly more important role in changes in precipitation than is the case for SAT and SST. For
precipitation, there is a large difference in $FCO_2$ over land compared to oceans, with mean values of 0.23 and 0.58 respectively; non-$CO_2$ forcing is much more important over land.

The largest increases in precipitation are generally driven by non-$CO_2$ forcing, seen over northern Africa and the Indian subcontinent (see Feng et al., 2022), Greenland, and parts of eastern and western Antarctica. Parts of northern Africa have $FCO_2$ values below 0.0, indicating that $CO_2$ is acting to limit this increase in precipitation. $FCO_2$ values below 0.0 are also
seen in central Australia, central North America and parts of the tropical Indian, Atlantic and Pacific Oceans, where the signals in precipitation anomaly are both positive and negative.

Non-$CO_2$ forcing is also dominant ($FCO_2$ 0.2-0.4) or highly dominant ($FCO_2$ 0.0-0.2) in some regions of precipitation decrease, including the tropical south Pacific, and the regions west of the maritime continent and off the eastern coast of North America.



There are also regions where $FCO_2$ is above 1.0 in parts of central and eastern Antarctica, the tropical Pacific, the Barents Sea,
and a small area in both the Bering Sea and the Arctic Ocean north of Alaska. These are mostly regions of small precipitation
increase, indicating that non-$CO_2$ forcing acts to decrease precipitation despite the overall increase.

Spatial changes appear predominantly driven by non-$CO_2$ forcing, whereas $CO_2$ forcing has a more muted and widespread
effect. The overall effect of $CO_2$ is an increase in global mean precipitation, although we see both increases and decreases in
precipitation regionally which appear attributable to non-$CO_2$ forcing such as changes in orography, ice sheets and/or
vegetation. That such local changes have a notable effect on the Pliocene precipitation anomaly may limit the degree to which
we can use the Pliocene as a precipitation analogue for our warmer future.

There is more uncertainty between models for $FCO_2$ on precipitation than for SAT and SST. Uncertainty is seen in regions of
both mixed forcing ($FCO_2$ 0.4-0.6) and dominant or highly dominant $CO_2$ forcing ($FCO_2$ 0.6-1.0). Regions predominantly
driven by non-$CO_2$ forcing ($FCO_2$ 0.0-0.4) show better agreement between models, suggesting that the impact of non-$CO_2$
forcing is more robustly represented in the PlioMIP2 ensemble than the impact of $CO_2$ on precipitation.

## 4. Discussion

### 4.1. FCO₂ method

The $FCO_2$ method has been validated by comparing outputs to the energy balance analysis and presents a great opportunity to
expand our understanding of climate drivers in the Pliocene and beyond.
We devised a novel method to quantitatively estimate the drivers of Pliocene SST and precipitation. This method can be applied
to other climate variables, with relative ease and little computational cost, and to other ensembles of models beyond PlioMIP2.
Aided by comparison to the energy balance analysis, the $FCO_2$ method provides a complete view of drivers of Pliocene climate
at both global and regional scales; in particular, contributions of $CO_2$ vs. non-$CO_2$ forcing for SAT, SST and precipitation on
local and regional scales are revealed. We also show how comparison to the energy balance analysis adds insight into feedbacks
and other such indirect effects of $CO_2$ forcing which the $FCO_2$ method does not capture.

This work has also highlighted the value and accuracy of using the $E^{400}$-$E^{280}$ and $Eoi^{400}$-$E^{400}$ SAT anomalies as an estimate for
$\Delta T_{gg\varepsilon}$ and the sum of the non-greenhouse gas components in energy balance analyses respectively. This shows that, while
exact information on the drivers of temperature still depend on the application of a more elaborated and computationally
expensive set of sensitivity simulations, a good degree of knowledge may be derived by applying a much smaller number of
simulations. This is not only more economic, but it may also increase the number of modelling groups that take part in future
model intercomparison studies of the kind that we have presented here. The $FCO_2$ method requiring a smaller number of
simulations compared to the energy balance analyses has allowed for a larger ensemble of models to be assessed than
previously in PlioMIP2.




The $FCO_2$ method also allows for an assessment of the uncertainty between models for the drivers of the different climate
parameters by comparing where there is/not consistent agreement on the forcing, i.e. whether $FCO_2 > 0.5$ or $FCO_2 < 0.5$ (Fig.
7).

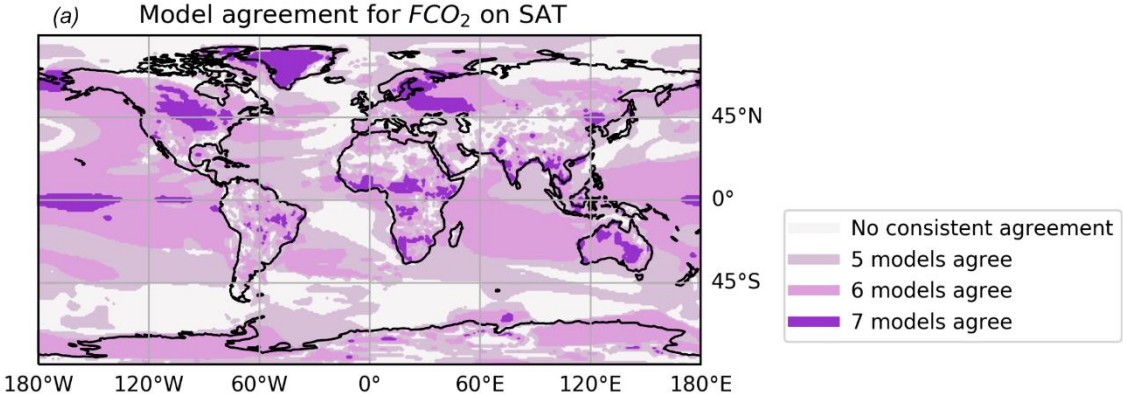

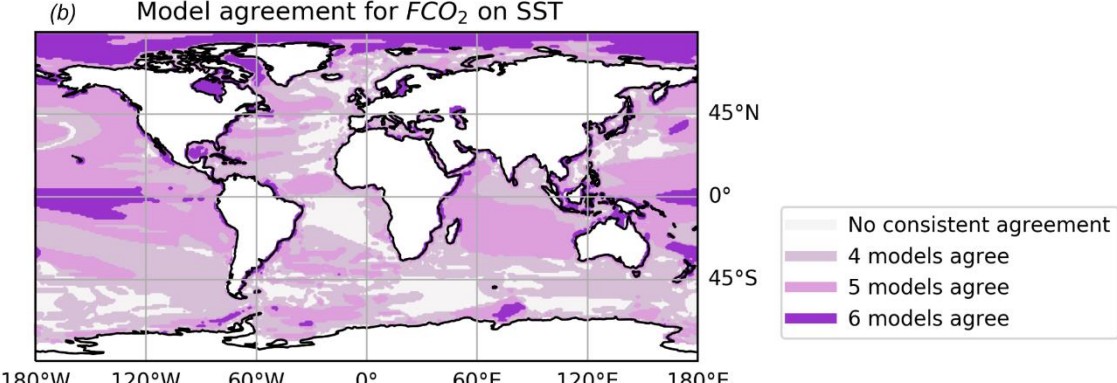

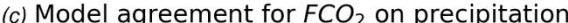

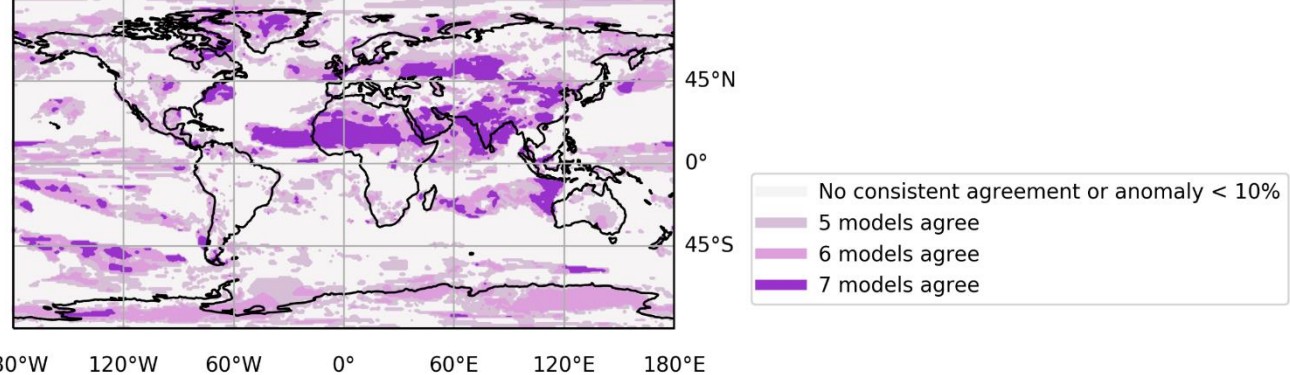

**Figure 7: The level of agreement between models included in the $FCO_2$ analysis, showing where models agree on the dominant
forcing shown by the $FCO_2$ method (i.e. whether $FCO_2 > 0.5$ or $FCO_2 < 0.5$). All seven models (CCSM4-UoT, CESM2, COSMOS,**





**HadCM3, IPSLCM5A2, MIROC4m and NorESM1-F) are included for the SAT and precipitation analysis; IPSLCM5A2 is excluded from the SST analysis as only ten model years of data were available hence a maximum of six models in agreement for SST. For precipitation, agreement is only assessed in regions where the Eoi$^{400}$-E$^{280}$ precipitation anomaly is greater than 10% for consistency.**

There is consistent agreement between five or more models on the dominant forcing of SAT over 74.8% of the Earth's surface (Fig. 7a), of SST over 46.5% of the ocean surface (Fig. 7b), and of precipitation over 66.8% of regions with an Eoi$^{400}$-E$^{280}$

anomaly greater than 10% (Fig. 7c). If the criteria for 'consistency' is extended to four or more models for SST – for which only six models are assessed – the area in agreement increases to 83.1%.

Though FCO$_2$ on precipitation is not the most consistent, in regions of agreement it is more common for all seven models to agree: all seven models agree on the dominant forcing in 13.8% of the area assessed for precipitation, compared to 4.6% for SAT. All six models agree on the dominant forcing for SST over 11.4% of the ocean surface.

**4.2. Drivers of Pliocene climate**

Using the FCO$_2$ method, CO$_2$ forcing was found to be the largest cause of SAT, SST and precipitation change in the Pliocene with global mean MMM FCO$_2$ values of 0.56, 0.56 and 0.51 respectively.

The percentage of SAT change predominantly driven by CO$_2$ using the FCO$_2$ method, 56%, is comparable to estimates from previous studies, including specific comparisons for HadCM3 (Lunt et al., 2012), CCSM4-UoT (Chandan and Peltier, 2018)

and COSMOS (Stepanek et al., 2020), as well as the PlioMIP1 ensemble (Hill et al., 2014).

Lunt et al. (2012) concluded that 48% of warmth simulated in HadCM3 was caused by CO$_2$ when the atmospheric concentration was set to 400 ppm, decreasing to 36% at 350 ppm and increasing to 61% at 450 ppm. Exploring the effect of different atmospheric CO$_2$ concentrations in this way would be possible using the FCO$_2$ method but is constrained by the experiments set out in the PlioMIP2 experimental design; further division of forcing factorisation experiments and/or more models

conducting these experiments (particularly the separated Eo$^{400}$ and Ei$^{400}$ experiments) may be a fruitful addition looking to PlioMIP3.

Using the FCO$_2$ method, 59% of the Pliocene SAT anomaly is caused by CO$_2$ in HadCM3 (global mean FCO$_2$ = 0.59; Fig. S2d). This is higher than the estimate of 48% in Lunt et al. (2012) but it is important to note the development in boundary conditions from PRISM2 (used in Lunt et al., 2012) and PRISM4 (used in PlioMIP2) which will account for some of the

difference, as well as the difference in methodology.

The percentage of warming predominantly caused by CO$_2$ using the FCO$_2$ method in CCSM4-UoT, 52% (global mean FCO$_2$ = 0.52; Fig. S2a), is also higher than the ~45% estimated in Chandan and Peltier (2018) using the nonlinear factorisation methodology of Lunt et al. (2012).

On the other hand, the FCO$_2$ method slightly underestimates the contribution of CO$_2$ in COSMOS compared to the full

factorisation in Stepanek et al. (2020). The global mean FCO$_2$ for COSMOS is found to be 0.64 (64% CO$_2$ contribution equivalent), compared to 66% in Stepanek et al. (2020). This may reflect the incorporation of some vegetation feedback in the





$E^{400}$-$E^{280}$ anomaly used to calculate $FCO_2$ given that COSMOS ran with dynamic vegetation, but additional simulations of COSMOS using prescribed vegetation would be needed to explore this further.

In validating the $FCO_2$ method, this paper has also presented the first energy balance results for a subgroup of models in the

PlioMIP2 ensemble. By using the same methodology as Hill et al. (2014) in the framework of PlioMIP1, our results based on PlioMIP2 experiments become directly comparable and similar trends are seen: greenhouse gas emissivity is dominant in driving warming in the tropics while all forcing components become important in the high-latitudes, with polar amplification particularly driven by clear-sky albedo. The relative dominance of $CO_2$ forcing in the low- and mid-latitudes compared to the high-latitudes is also seen in the $FCO_2$ results.

We find notable variation of results based on the $FCO_2$ method between individual climate models, though the level of variation is consistent between the three climate variables assessed. Despite having the highest ECS value in the PlioMIP2 ensemble (5.3°C; Gettelman et al., 2019; Haywood et al., 2020), CESM2 has the lowest $FCO_2$ for all three variables at 0.40 for SAT and SST, and 0.39 for precipitation. This further highlights the sensitivity of CESM2 to all changes in boundary conditions, not just $CO_2$ (Feng et al., 2020).

The model with the highest global mean $FCO_2$ differs between variables. NorESM1-F has the highest $FCO_2$ on SAT at 0.70, while COSMOS has the highest $FCO_2$ on SST and precipitation at 0.76 and 0.69, respectively. NorESM1-F has the lowest ECS value in the PlioMIP2 ensemble (2.3°C), but COSMOS has the third highest (4.7°C; Haywood et al., 2020). Though it might seem intuitive that models with a higher ECS would also have a higher $FCO_2$, the relationship between $FCO_2$ and climate sensitivity can be better described by the ESS to ECS ratio which captures the relatively short-term influence of $CO_2$ compared

to longer-term responses of the Earth System (Fig. 8). Perhaps an artefact of the reduced sample size (six models compared to seven), the ESS to ECS ratio correlates best with the global mean $FCO_2$ on SST ($R^2 = 0.71$), followed by SAT ($R^2 = 0.50$) and precipitation ($R^2 = 0.41$). This relationship would be better explored with a greater sample size, again reinforcing the usefulness of model groups completing the forcing factorisation experiments ahead of PlioMIP3.





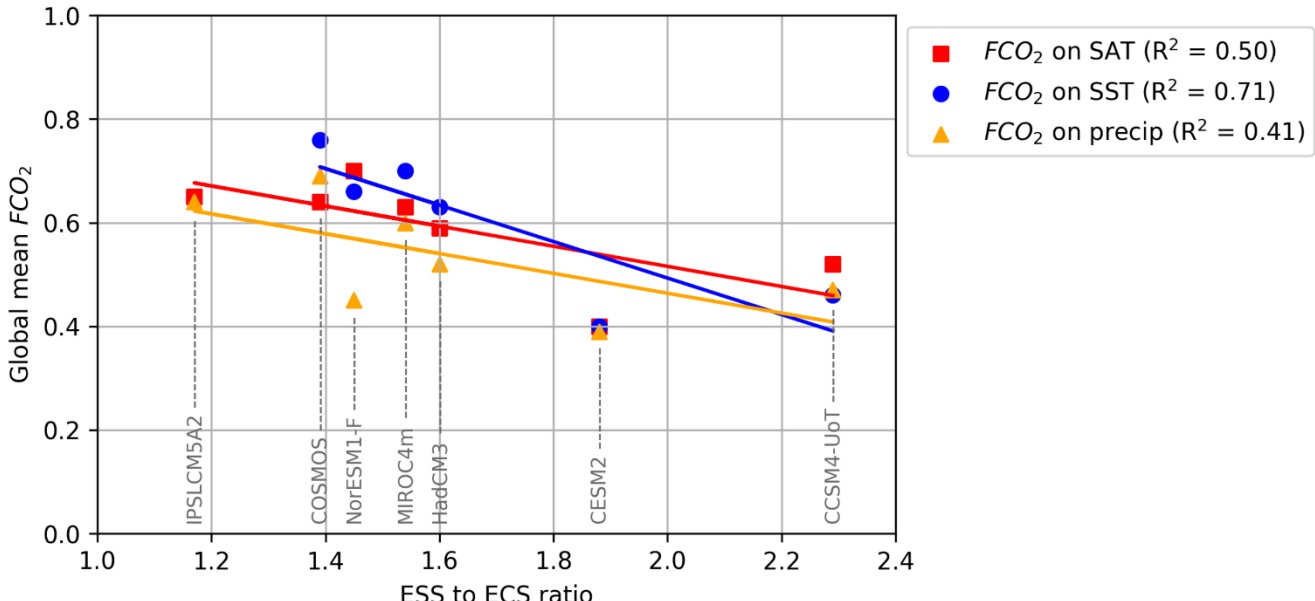

**Figure 8: The relationship between the ESS to ECS ratio and global mean $FCO_2$. IPSLCM5A2 is excluded from the SST analysis due to limited data availability.**

**4.3. The Pliocene as an analogue for the future?**

A significant motivation behind studying the Pliocene is its use as a potential palaeoclimate analogue for the near-term future. If the Pliocene is to be an accurate and useful analogue for the future, it stands that the drivers of its climate should also be analogous to those driving current anthropogenic climate change alongside its large-scale climate features.

The $FCO_2$ method allows us to answer the question of how analogous the drivers of Pliocene climate are to those of the near-term future in more detail than has been possible previously. It also allows us to consider this question in terms of SST and precipitation change for the first time.

Current warming is predominantly driven by anthropogenic greenhouse gas emissions (Eyring et al., 2021). The $FCO_2$ results presented here show that, although $CO_2$ was the most important forcing in the Pliocene, it drove only 56% of SAT and SST change and 51% of precipitation change in the ensemble of PlioMIP2 models considered in this study. Therefore, 44% of SAT and SST change and 49% of precipitation change was driven by non-$CO_2$ forcing.

While we are already experiencing some shifts towards a Pliocene-like state for some of these non-$CO_2$ components – such as the greening of the Arctic (e.g. Myers-Smith et al., 2020) – other changes will take longer to fully materialise as the system equilibrates to higher levels of anthropogenic $CO_2$ forcing, with implications on the accuracy and utility of the Pliocene as a palaeoclimate analogue for near-term future climate. Regions of high $FCO_2$ in the Pliocene are likely to be more analogous for the immediate and near-term future for as long as atmospheric $CO_2$ concentration remains similar to Pliocene levels (~400





ppm), whereas regions of lower $FCO_2$ may become more analogous in the longer-term future as the full, equilibrated effects of changes to ice sheets and vegetation are experienced.

This raises two important points. The first highlights the importance of understanding the broader Earth System feedbacks of an atmospheric $CO_2$ concentration similar-to-modern, particularly as anthropogenic greenhouse gas emissions continue to increase (Dhakal et al., 2022) with the likelihood of soon moving beyond Pliocene levels (~400 ppm; Meinshausen et al., 2020). The $E^{400}$-$E^{280}$ SAT anomaly shows that, for the subgroup of seven PlioMIP2 models assessed here, $CO_2$ forcing alone was responsible for 1.8°C of the total 3.2°C increase seen in the $Eoi^{400}$-$E^{280}$ global mean SAT anomaly. We have experienced

around 1.1°C of warming relative to the PI with an atmospheric $CO_2$ concentration of around 410 ppm (Gulev et al., 2021). The Pliocene – being around 3°C warmer than the PI in quasi-equilibrium with a $CO_2$ concentration ~400 ppm (e.g. Haywood et al., 2020) – shows that more warming is to come as the system equilibrates with the anthropogenic greenhouse gas forcing that has already been emitted, even if greenhouse gas emissions were to stop immediately.

The second point highlights the need to define what we mean by palaeoclimate analogue in the situation of our research. This

should include consideration of the climate variable/s, region/s and time frame/s of interest (including whether the system is in a transient or equilibrium state, with implications for modes of variability (e.g. Bonan et al., 2022)), as well as the level of accuracy deemed to be 'analogous'. Our results also highlight the need to consider the nature of the climate forcing.

Burke et al. (2018) explore the spatial and temporal variations of past warm periods as analogues for different potential climate futures by comparing six geohistorical periods (PI, Historical, Holocene, Last Interglacial, Pliocene and Eocene) to

Representative Concentration Pathway (RCP) 4.5 and RCP8.5. They find the Pliocene to be the best analogue for our near-term future under RCP4.5, though just because the Pliocene is one of the best palaeoclimate analogues does not necessarily mean that it is a perfect analogue without constraints or limitations.

Future work could expand the work of Burke et al. (2018) using the $FCO_2$ method to incorporate additional climate variables, which would also allow for discussion on the analogous nature of the drivers of these variables.

The results presented here highlight that, though there may be similarities in large-scale features of Pliocene and near-term future climate, the drivers of these features may be less similar or analogous and drawing any such conclusions must be done so with caution and account for the significant contributions of non-analogous forcings.

## 5. Summary and future work

We have introduced a novel method for assessing the influence of different forcing factors in the Pliocene. The $FCO_2$ method

only requires a small subset of forcing factorisation experiments of PlioMIP2 and can be applied to multiple climate variables, and to a large ensemble of models with little computational complexity and cost. We have validated the $FCO_2$ method by comparing the results for SAT to an energy balance analysis using the methodology of Hill et al. (2014), which was originally used to assess the drivers of warming in the PlioMIP1 ensemble.

For the first time, we have quantitatively estimated the effect of $CO_2$ forcing on Pliocene SST and precipitation. $CO_2$ is found

to be the most important forcing of global mean SAT, SST and precipitation with global mean $FCO_2$ values of 0.56 (individual model range 0.40-0.70), 0.56 (individual model range 0.40-0.76) and 0.51 (individual model range 0.39-0.69) respectively. Though $CO_2$ is the most important forcing, there remains significant contributions from non-$CO_2$ forcing and such changes in orography, ice sheets and/or vegetation are found to have a greater impact in driving regional spatial changes. The influence of these non-$CO_2$ forcings must not be overlooked, particularly in the context of using the Pliocene as an analogue for the near-

term future.

Outputs from the $FCO_2$ method also provide new insights relevant to the palaeo-data community which could aid the interpretation of proxy data and data-model comparison efforts, as well as inform estimates of climate sensitivity. These insights will be explored in a future paper. The $FCO_2$ method shows us which regions of the world are most (and least) influenced by $CO_2$ forcing, with direct implications on the interpretation of proxy data at these sites and any biases they may

present. Additionally, we can also use the outputs from the $FCO_2$ method to suggest regions from which additional proxy data would be useful to further refine our interpretation of Pliocene climate, such as where there is uncertainty between models.

As we look towards the planning of PlioMIP3, our work clearly highlights the usefulness and importance of including forcing factorisation experiments that can provide us with a more detailed view of the drivers of Pliocene climate, with direct relevance to the discussion on using the Pliocene as an analogue for our warmer future.

**Data availability**

The data required to produce the results in this paper is available in the Supplement.

**Author contribution**

LEB, AMH, JCT, AMD and DJH prepared the paper with contributions from all co-authors. AAO, WLC, DC, RF, SJH, XL, WRP, NT, CS and ZZ provided PlioMIP2 experiments run with the individual models.

**Competing interests**

Some authors are members of the editorial board of Climate of the Past. The peer-review process was guided by an independent editor, and the authors have also no other competing interests to declare.

**Acknowledgements**

For the purpose of open access, the author has applied a Creative Commons Attribution (CC BY) licence to any Author
Accepted Manuscript version arising. Lauren E. Burton acknowledges that this work was supported by the Leeds-York-Hull



Natural Environment Research Council (NERC) Doctoral Training Partnership (DTP) Panorama under grant NE/S007458/1. Research by Wing-Le Chan and Ayako Abe-Ouchi was supported by JSPS Kakenhi grant no. 17H06104 (Japan) and MEXT Kakenhi grant no. 17H06323 (Japan). Christian Stepanek acknowledges support at AWI by the Helmholtz Climate Initiative REKLIM and the research program PACES-II of the Helmholtz Association. W. Richard Peltier and Deepak Chandan were

supported by Canadian NSERC Discovery Grant A9627, and they wish to acknowledge the support of SciNet HPC Consortium for providing computing facilities. SciNet is funded by the Canada Foundation for Innovation under the auspices of Compute Canada, the Government of Ontario, the Ontario Research Fund – Research Excellence, and the University of Toronto. Ning Tan acknowledges support from National Natural Science Foundation of China program 41907371. Stephen J. Hunter was supported by the FP7 Ideas: European Research Council (grant no. PLIO-ESS, 278636). The CESM2 simulations are

performed with high-performance computing support from Cheyenne (doi:10.5065/D6RX99HX) provided by NCAR's Computational and Information Systems Laboratory, sponsored by the National Science Foundation. Ran Feng acknowledges support from National Science Foundation (No. 2103055). The NorESM simulations benefitted from resources provided by UNINETT Sigma2 – the National Infrastructure for High Performance Computing and Data Storage in Norway.

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
