# Peer review of "On the climatic influence of CO2 forcing in the Pliocene"

_Climate of the Past, 2022_

## Author Response (AR1)

**On the climatic influence of $CO_2$ in the Pliocene**

**Response to Anonymous Referee 1 (RC1)**

The authors thank Anonymous Referee #1 for their comments and feedback. Answers to specific questions are provided below and revisions are indicated in the revised manuscript.

*I have one comment on the response of precipitation. In P15 L352, "The largest increases in precipitation are generally driven by non-CO2 forcing, seen over northern Africa and the Indian subcontinent, ....", the responses in the high-latitudes may be related to the opposed feedbacks due to the changes of topography and ice-sheet. But what are the non-CO2 forcing for the low latitudes? Given that most models use the prescribed vegetation. It would be good to separately discuss the precipitation response in high and low latitudes. Some discussions may need in particular for the eye-catching precipitation pattern in north Africa as shown in Fig6b and Fig1d.*

The increase in precipitation noted here is a robust signal seen in the broader PlioMIP2 ensemble. The $Eoi^{400}$ simulation considers changes in boundary conditions besides just $CO_2$, including changes to orography and ice sheets, as noted in the original manuscript. The changes in ice sheet volume and extent lead to changes in atmospheric circulation, including changes in the Hadley cell and consequently to precipitation in northern Africa. Such changes to the Hadley cell are reported in both PlioMIP1 and PlioMIP2.

Furthermore, it is important to note that precipitation change is shown as a percentage relative to the pre-Industrial in Figure 1d and Figure 6b, meaning that these changes may appear to be large features when the absolute change in precipitation is low compared to other regions.

A sentence has been added in the revised manuscript to clarify the impact of ice sheet changes in precipitation in low-latitude regions due to changes in atmospheric circulation.

*P3-4, Fig1, Are the CMIP6 future scenarios using the same models as those 16 models in PlioMIP2?*

The models included in PlioMIP2 are not all included in CMIP6. A sentence has been added to the caption of Figure 1 to clarify this.

*P3, L70, the text mention "There are also comparable spatial patterns of climate anomalies ... in the form of polar amplification", one can notice that Pliocene has polar amplification in both the Arctic and Antarctic, but SSP2-4.5 only show amplification in the Arctic but not in the Antarctic. One also can notice a dramatic difference between Pliocene and SSP2-4.5 in the precipitation over North Africa including the middle east. The reasons for these differences are mentioned in the late part of the manuscript, would be good to point out this obvious difference here when the figures are presented.*

These differences can be explained by the effects of Pliocene boundary conditions which are not comparable to those for future scenarios. The differences in polar amplification and precipitation in the noted regions can largely be attributed to the differences in the Antarctic ice sheet in the Pliocene (PlioMIP2 $Eoi^{400}$ simulation) and the future. Feedbacks arising from the smaller extent of the Antarctic ice sheet in the Pliocene drives polar amplification in this region. The smaller ice sheet also affects the equator-to-pole temperature gradient, changing atmospheric circulation and leading to a broadening of the Hadley cell. As noted above in response to a previous comment, this is a robust result in both PlioMIP1 and PlioMIP2.

A sentence has been added in the revised manuscript to clarify the causes of these differences.

*P9, L211, in equation (4), it should be SW\*(1-α) not SW-(1-α)*

This equation has been corrected in the revised manuscript.

**Response to Anonymous Referee 2 (RC2)**

The authors thank Anonymous Referee #2 for their comments and feedback. The revisions made are indicated in the revised manuscript.